# Investigation of the Néel phase of the frustrated Heisenberg antiferromagnet by differentiable symmetric tensor networks

**Juraj Hasik[1]**$^\star$**, Didier Poilblanc[1] and Federico Becca[2]**

**1** Laboratoire de Physique Théorique UMR5152, C.N.R.S. and Université de Toulouse,
118 rte de Narbonne, 31062 Toulouse, FRANCE
**2** University of Trieste, Department of Physics, Strada Costiera 11, 34151 Trieste, ITALY

$\star$ juraj.hasik@irsamc.ups-tlse.fr

## Abstract

The recent progress in the optimization of two-dimensional tensor networks [H.-J. Liao, J.-G. Liu, L. Wang, and T. Xiang, Phys. Rev. X 9, 031041 (2019)] based on automatic differentiation opened the way towards precise and fast optimization of such states and, in particular, infinite projected entangled-pair states (iPEPS) that constitute a generic-purpose *Ansatz* for lattice problems governed by local Hamiltonians. In this work, we perform an extensive study of a paradigmatic model of frustrated magnetism, the $J_1-J_2$ Heisenberg antiferromagnet on the square lattice. By using advances in both optimization and subsequent data analysis, through finite correlation-length scaling, we report accurate estimations of the magnetization curve in the Néel phase for $J_2/J_1 \leq 0.45$. The unrestricted iPEPS simulations reveal an $U(1)$ symmetric structure, which we identify and impose on tensors, resulting in a clean and consistent picture of antiferromagnetic order vanishing at the phase transition with a quantum paramagnet at $J_2/J_1 \approx 0.46(1)$. The present methodology can be extended beyond this model to study generic order-to-disorder transitions in magnetic systems.

 Check for updates

# 1  Introduction

The spin-$S$ antiferromagnet, with isotropic coupling $J_1$ between nearest-neighbor spins located on the sites of a square lattice, represents one of the most paradigmatic models of quantum magnetism. At zero temperature, the system develops long-range antiferromagnetic (Néel) order for any value of $S$: while for $S \geq 1$ there are analytical arguments [1,2], for the extreme quantum case with $S = 1/2$, this has been numerically proven thanks to quantum Monte Carlo simulations on large systems [3–5]. Instead, any finite temperature will restore spin rotation symmetry, in agreement with the Mermin-Wagner theorem [6]. A magnetically disordered ground state may be also achieved by including further super-exchange couplings, most notably a next-nearest-neighbor interaction $J_2$, which destabilizes the Néel order driving towards a quantum phase transition. In this respect, much effort has been spent to understand the ground-state properties of the $J_1 - J_2$ model defined by:

$$\mathcal{H} = J_1 \sum_{\langle i,j \rangle} \mathbf{S}_i \cdot \mathbf{S}_j + J_2 \sum_{\langle\langle i,j \rangle\rangle} \mathbf{S}_i \cdot \mathbf{S}_j \,. \tag{1}$$

Here, $\langle \ldots \rangle$ and $\langle\langle \ldots \rangle\rangle$ stand for nearest-neighbor and next-nearest-neighbor sites on the square lattice, respectively; $\mathbf{S}_i = (S_i^x, S_i^y, S_i^z)$ represents the spin-1/2 operator on the site $i$. Both the spin-spin interactions are taken positive.

In the presence of finite $J_2$ a severe sign problem is present (especially in the local basis with $z$-component defined on each site), which prohibits quantum Monte Carlo algorithms from assessing large system sizes. Over the last three decades several alternative methods have been introduced and kept improving, e.g., exact diagonalizations [7–9], series expansions [10,11], coupled-cluster [12,13] and cluster mean-field approaches [14,15], density-matrix renormalization group (DMRG) [16–18], functional-renormalization group (fRG) [19], and variational Monte Carlo (VMC) approaches [20–23]. At present, there is a strong evidence that the ground state has no magnetic order from $J_2/J_1 \approx 0.5$ to $J_2/J_1 \approx 0.6$. However, the investigations of the nature of the paramagnetic phase have led to contradicting results, supporting the existence of a valence-bond solid (with either columnar or plaquette order) [7–12] or a spin liquid (either gapped or gapless) [13,14,16,19,20], or even both [17,18,21–23]. One important aspect emerging in the latest calculations is the existence of a *continuous* quantum phase transition between the antiferromagnetic and the paramagnetic phases for $J_2/J_1 \approx 0.5$, where the staggered magnetization (hereafter named simply "magnetization") goes to zero.

Recently, borrowing concepts from quantum information, tensor-network methods have been introduced [24–26]. In one dimension, the so-called matrix-product states (MPS) offer a convenient and elegant rephrasing of previous DMRG ideas. MPS evolved into the method of choice and provide very accurate approximations of the exact ground-state properties. Generalizations in two dimensions are more problematic. The prominent example, projected entangled-pair states (PEPS), provide the correct entanglement structure of most quantum ground states of local spin Hamiltonians [27], however, they suffer from a steep scaling of computational effort when enlarging the system size. For this reason, their application has been limited to ladder geometries with small number of legs [28,29] and finite 2D clusters with open boundary and up to $\approx 200 - 300$ sites [30]. In order to overcome this computational barrier

and avoid boundary effects, algorithms that work directly in the thermodynamic limit (dubbed iPEPS) have been introduced and developed [31, 32]: here, only a small number of tensors is explicitly considered and embedded into an environment that is self-consistently obtained (e.g., within the so-called corner-transfer matrix approaches [33] or channels [34]). The size of these tensors, and in turn the number of variational parameters of the wave function, is characterized by the so-called bond dimension $D$. The iPEPS are systematically improved by enlarging the bond dimension, accounting for increasingly entangled states.

In recent years, iPEPS have been applied to assess the nature of the ground state of the $J_1-J_2$ model, mainly focusing on the highly-frustrated regime $J_2/J_1 \approx 0.5$ [35–37]. However, these attempts were not completely satisfactory, since they either used a simplified tensor structure, limited to the description of paramagnets, or suffer from optimization problems, arising in methods that are not fully satisfactory and consistent (e.g., the so-called simple and full update [31, 38]). In this respect, a breakthrough in the field has been achieved by performing the tensor optimization using the ideas of algorithmic differentiation, or better the *adjoint algorithmic differentiation* (AAD) technique, which allow a very efficient optimization even in the presence of large number of parameters [39]. Here, Liao and collaborators limited their application to the unfrustrated Heisenberg model (with $J_2 = 0$), showing that extremely accurate and completely stable results may be obtained for both the ground-state energy and magnetization.

Even though PEPS (and iPEPS) *Ansätze* are designed to describe both gapped and gapless states (following the entanglement entropy's area law, up to additive corrections), it remains an open question whether generic optimization can reliably reproduce highly-entangled ground states, as the ones that are possibly emerging in the frustrated regime $J_2/J_1 \approx 0.5$ [20–23]. Therefore, in this work, we do not directly address the question of the nature of the magnetically disordered phase; instead, we focus our attention to the magnetically ordered phase with $J_2/J_1 \leq 0.45$ and perform an accurate determination of the magnetization curve as a function of the frustrating ratio. In addition to its conceptual importance, the problem of the disappearance of antiferromagnetic order under increasing frustration offers a stringent test to most numerical methods, in general, and to tensor network methods, in particular. To this end, we apply the same ideas of AAD to optimize the iPEPS *Ansatz* for the $J_1-J_2$ model of Eq. (1). Importantly, unlike the previously proposed gradient-based optimizations [34, 40], the AAD can be effortlessly extended beyond nearest-neighbour Hamiltonians. The energy and magnetization are obtained for different values of the bond dimension $D$, from 2 up to 7. Then, the estimates for $D \to \infty$ are obtained for each frustration ratio $J_2/J_1$. Note however that, this is *not* realized by a crude extrapolation in $1/D$ (for which the results for different values of $D$ are considerably scattered) but, instead, by performing a correlation-length extrapolation, which is motivated by the finite-size scaling analysis that is well established in the Néel phase, as recently proposed in Refs. [41, 42]. Despite the fact that this mode of extrapolation requires the calculation of the correlation length $\xi$, which may not be as accurate as other thermodynamic quantities (e.g., energy and magnetization), it has been shown to give remarkably good results for the unfrustrated Heisenberg model. In fact, as we have mentioned earlier, even though iPEPS can describe certain gapless phases, their generic optimization instead leads to states with finite correlation lengths. e.g., in the Néel phase, and the bond dimension $D$ turn out not to be the correct object to quantify this aspect. As we will show, also in the presence of frustration, the analysis based on the correlation length gives reliable thermodynamic estimates, even though no exact results are available. Our calculations are compatible with a vanishing magnetization for $J_2/J_1 \approx 0.45$, which is in close agreement with recent calculations [13, 18, 20–23] and give a reference for future investigations.

The paper is organized as follows: in section 2, we will describe the iPEPS method; in section 3, we present the results; in section 4, we finally draw our conclusions and discuss the

perspectives.

## 2  iPEPS *Ansatz* **and its optimization**

We parametrize the state by a single real tensor

$$a^s_{uldr} = \quad (2)$$

with a physical index $s = \uparrow, \downarrow$ labeling the standard $S^z$ basis of the local physical Hilbert space and auxiliary (or virtual) indices $u, l, d, r$ of bond dimension $D$ (by convention running from 0 to $D-1$ here). The physical wave function is then obtained by tiling the infinite square lattice with tensor $a$ and tracing over all auxiliary indices

$$\psi(a) = \sum_{\{s\}} c(a)_{\{s\}} |\{s\}\rangle$$

$$c(a)_{\{s\}} := \mathrm{Tr}_{aux}(a^{s_0} a^{s_1} a^{s_2} \dots) = \quad (3)$$

The tensor $a$ is chosen (and constructed such as) to be invariant under a number of symmetries. First of all, it belongs to the $A_1$ irreducible representation of the $C_{4v}$ point group, thus enforcing all the spatial symmetries of the square lattice on the iPEPS. The antiferromagnetic correlations are incorporated in the ansatz by unitaries $-i\sigma^y$, which rotate the physical $S^z$ basis at every site of one sublattice. We absorb these unitaries into observables leaving the definition of the wave function untouched [see Eq. (12)].

Secondly, the tensor $a$ also possesses a further structure by requiring certain transformation properties under the action of $U(1)$ group (see below). Such choice is motivated by the remaining $U(1)$ symmetry in the ordered phase, which manifests itself as equivalence between different magnetizations connected by transverse (Goldstone) modes. As defined below, $U(1)$ tensor classes are defined by assigning specific "charges" to the virtual and physical degrees of freedom.

When considering $A_1$- and $U(1)$-symmetric states, the tensor $a = a(\vec{\lambda})$ is taken to be a linear combination of (fixed) elementary tensors $\{t_0, t_1, \dots\}$ (named a tensor "class") such that

$$a(\vec{\lambda}) = \sum_i \lambda_i t_i, \quad (4)$$

with coefficients $\vec{\lambda}$ being the variational parameters. The elementary tensors $\{t_0, t_1, \dots\}$ are different representatives of the $A_1$ irreducible representation for some choice of the $U(1)$ charges.

Given an iPEPS defined by tensor $a$, the evaluation of any observable $\mathcal{O}$ amounts to a contraction of infinite double-layer network composed of tensors $a$ together with the tensor representation of $\mathcal{O}$. Such tensor network is the diagrammatic equivalent of usual expression $\langle \mathcal{O} \rangle = \langle \psi(a) | \mathcal{O} | \psi(a) \rangle$. A central aspect of iPEPS method is an approximate contraction of such networks. In this work, we realize them by finding the so-called environment tensors $C$ and $T$ of dimension $\chi$, dubbed environment dimension, by the means of corner-transfer matrix (CTM) procedure [33]. These tensors compress the parts of the original infinite network in approximate but finite-dimensional objects. Afterwards, the desired reduced density matrices can be constructed from $C$ and $T$, together with the on-site tensor $a$. Ultimately, the exact

Figure 1: Definition of reduced-density matrices necessary for evaluating the energy per site of the $J_1 - J_2$ model over single-site iPEPS with $C_{4v}$ symmetry. (a) Double-layer tensors with contracted and uncontracted physical indices. (b) Infinite tensor network corresponding to the next-nearest-neighbour $\rho^{(NNN)}$ as approximated by $\rho_\chi^{(NNN)}$ in the finite network with $C$ and $T$ tensors resulting from CTM. (c) Finite-network approximation of nearest-neighbour $\rho_\chi^{(NN)}$ within the same $2 \times 2$ cluster.

value of any observable is recovered taking $\chi \to \infty$, which we extrapolate from the data for increasingly large $\chi$.

The optimization of the tensor $a$ (or equivalently in the $U(1)$-symmetric approach, of the parameters $\vec{\lambda}$) is carried out using standard gradient-based method L-BFGS supplemented with backtracking linesearch. The gradients are evaluated by AAD, which back-propagates the gradient through the whole process of energy evaluation for fixed $\chi$ [39]: Starting with a given CTM, followed by assembling the reduced-density matrices from converged $C, T$ tensors and finally evaluating the spin-spin interaction between nearest and next-nearest neighbors.

## 2.1 Extracting the relevant $U(1)$ charges

For small enough frustration, in the Néel phase, the unconstrained optimization of tensor $a$ leading to correct $U(1)$-symmetric iPEPS would be a desirable outcome. Under circumstances, AAD optimization can arrive at an almost $U(1)$-symmetric tensor $\tilde{a}$. In such case, a direct and robust evidence can be seen in the nearly degenerate pairs of leading eigenvalues of the transfer matrix. Importantly, such iPEPS states provide an unbiased information about the energetically favourable $U(1)$-charge structure of tensor $a$. We are concerned with inferring these charges from the elements of tensor $\tilde{a}$. Obtaining the correct charge assignment of the smallest $D$ tensors allows (i) to perform an efficient variational optimization over a greatly reduced number of parameters $\vec{\lambda}$, (ii) to obtain truly $U(1)$-symmetric environments via CTM and, finally, (iii) to predict the correct charge content of higher-$D$ $a$ tensors and, hence, enable to perform (i) and (ii) for larger $D$.

Before describing how to achieve the goal of obtaining the charges from the almost sym-

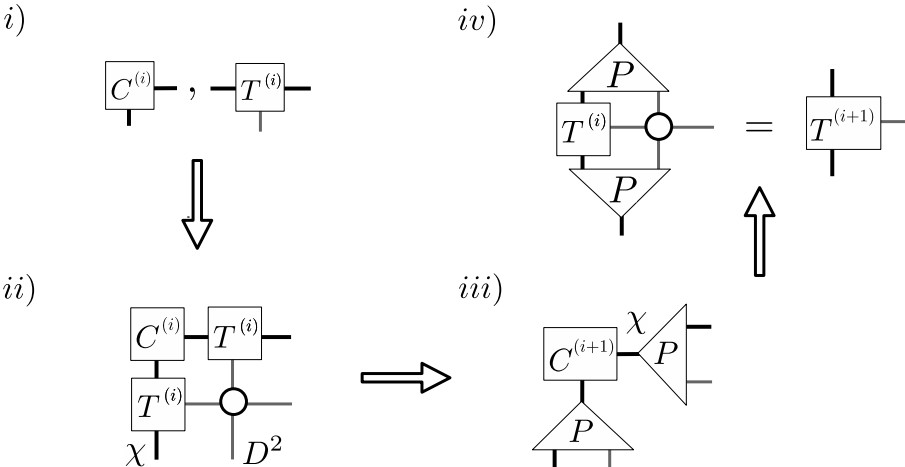

Figure 2: Key steps of the CTM algorithm for single-site iPEPS with $C_{4v}$ symmetry. (i) Initial tensors at iteration $i$: $\{C^{(i)}, T^{(i)}\}$. (ii) Construction of enlarged corner and its reshaping into matrix of dimensions $D^2\chi \times D^2\chi$. (iii) Symmetric eigenvalue decomposition of enlarged corner and truncation down to leading $\chi$ eigenpairs by magnitude of the eigenvalues. Truncation is always done at the boundaries between degenerate eigenvalues (see text). (iv) Absorption and truncation with isometry $P$ from step (ii) for half-row/-column tensor $T$.

metric $\tilde{a}$ tensor, let us first briefly review the expected properties of the resulting $U(1)$-symmetric $a$ tensor. In practice, one has to assign $U(1)$ charges $\vec{u} = (u^\uparrow, u^\downarrow)$ and $\vec{v} = (v_0, \ldots, v_{D-1})$ (which could be rational numbers) to the two physical spin-1/2 components and the $D$ virtual degrees of freedom on each of the four auxiliary indices. Without loss of generality we take them to be integers. Notice that in order to preserve $C_{4v}$ symmetry the same charges $\vec{v}$ are taken on each of four auxiliary legs of the tensor. The action of an element $g \in U(1)$ on $a$ is given by:

$$a^s_{uldr} \to (ga)^s_{uldr} = a^{s'}_{u'l'd'r'} U^{ss'} V_{uu'} V_{ll'} V_{dd'} V_{rr'}, \tag{5}$$

where $U$ and $V$ are diagonal matrices depending on $g$, and, in order to preserve $C_{4v}$ symmetry, all auxiliary indices are transformed by the same $V$:

$$U^{ss'} = e^{i\theta_g u^s} \delta^{ss'}, \tag{6}$$

$$V_{\gamma\gamma'} = e^{i\theta_g v_\gamma} \delta_{\gamma\gamma'}, \tag{7}$$

with the phase $\theta_g \in \mathbb{R}$ and $\gamma = 0, \ldots, D-1$. Therefore, the non-zero elements of the tensor $a$ transform according to

$$(ga)^s_{uldr} = a^s_{uldr} e^{i\theta_g (u^s + v_u + v_l + v_d + v_r)}. \tag{8}$$

In this language, the $U(1)$ invariance is realized by simply enforcing a selection rule for the non-zero tensor elements $a^s_{uldr}$, which should exhibit a local charge conservation

$$u^s + v_u + v_l + v_d + v_r = N. \tag{9}$$

Notice that there is some freedom in the definition of the charges since shifts like $u^s \to u^s + \alpha$, $v_\sigma \to v_\sigma + \beta$, and $N \to N + \alpha + 4\beta$, with $\alpha$ and $\beta \in \mathbb{Z}$, leave Eq. (9) invariant.

Hence, Eq. (9) implies that $a$ is indeed invariant up to global phase under the action of $U(1)$. Once the relevant $U(1)$ charges $\vec{u}$ and $\vec{v}$ are known (see below), practically, Eq. (9) is used in the construction of the elementary tensors $\{t_0, t_1, \ldots\}$ by filtering out their non-zero elements.

Table 1: $U(1)$ charges as inferred from unrestricted simulations with bond dimensions $D = 2, \ldots, 7$. Predictions of the charges for $D = 8$ and $9$ are also shown. Note that the ordering of the $v_\alpha$ charges is arbitrary and the gauge freedom has been fixed by taking $N = 1$. The last column shows the number of elementary tensors $t_i$.

| $D$ | $[u_\uparrow, u_\downarrow, v_0, v_1, \cdots, v_{D-1}]$ | number of tensors |
|---|---|---|
| 2 | $[1, -1, 0, 2]$ | 2 |
| 3 | $[1, -1, 0, 2, 0]$ | 12 |
| 4 | $[1, -1, 0, 2, -2, 0]$ | 25 |
| 5 | $[1, -1, 0, 2, -2, 0, 2]$ | 52 |
| 6 | $[1, -1, 0, 2, -2, 0, 2, -2]$ | 93 |
| 7 | $[1, -1, 0, 2, -2, 0, 2, -2, 2]$ | 165 |
| 8 | $[1, -1, 0, 2, -2, 0, 2, -2, 0, 2]$ | 294 |
| 9 | $[1, -1, 0, 2, -2, 0, 2, -2, 0, 2, -2]$ | 426 |

Let us now describe how to infer the charges from an unrestricted tensor optimization that has produced an almost symmetric on-site tensor $\tilde{a}$, with bond dimension $D$. To identify the dominant (at least for small $D$) $U(1)$-symmetric component of $\tilde{a}$, and then ultimately derive the hidden $U(1)$ charges, we have to first perform a higher-order singular value decomposition of $\tilde{a}$:

$$\tilde{a}^s_{uldr} = Z^{ss'} Y_{uu'} Y_{ll'} Y_{dd'} Y_{rr'} c^{s'}_{u'l'd'r'}, \tag{10}$$

with unitary matrices $Z$, $Y$, and the so-called core tensor $c$. The same unitary $Y$ is associated to different auxiliary legs due to the enforced $C_{4v}$ symmetry. The core tensor $c$ plays an analogous role to singular values in standard singular value decomposition of a matrix. The untruncated core tensor $c$ by itself defines a physically equivalent iPEPS to the one given by $\tilde{a}$. A good lower-rank approximation of $\tilde{a}$ can be obtained by truncation of the smallest elements of the core tensor $c$. The basic premise, supported by nearly degenerate transfer matrix spectrum for small $D$, is that the relative magnitude of symmetry-breaking elements of tensor $c$ is small. Therefore, we assume that the largest elements of tensor $c$ respect the $U(1)$-symmetry constrain associated to an unknown set of charges $\vec{u}$ and $\vec{v}$.

For the last step in identifying the charges, we re-formulate the problem in terms of linear algebra. First, taking a set of $n$ largest tensor elements (modulo $C_{4v}$ symmetry), and writing down Eq. (9) for each of them will result in a set of $n$ coupled linear equations (with integer coefficients) of the $D + 2$ unknown charges. Whenever $n > D + 2$, the linear system becomes over-complete and, increasing $n$ still allows the same solution for the charges, unless $n$ is taken too large so that (small) non-zero tensor elements breaking $U(1)$-symmetry are included. To solve this linear problem it is convenient to recast the constraints into a $n \times (D+2)$ matrix. The matrix, containing integer matrix elements, is obtained by simply counting the total number of charges of each type $\gamma$ and $s$ on the virtual and physical legs, respectively. More precisely, we define vectors $\vec{n}(c^s_{uldr})$ of integer coordinates that count the number of times specific *index value* appears among the indices of a given tensor element. Expressing each individual element constraint (9) as $\vec{n}(c^s_{uldr}) \cdot (\vec{u}, \vec{v}) = N$ and recasting them into matrix form, the linear system can be written in a compact fashion as $M \cdot (\vec{u}, \vec{v}) = \vec{N}$.

To be explicit, let us consider the case $D = 3$ for which all charges can be obtained using

only the $n = D + 2 = 5$ largest tensor elements of tensor $c$:

$$
\begin{matrix}
\vec{n}(c^{\uparrow}_{0000}) & \rightarrow \\
\vec{n}(c^{\downarrow}_{0001}) & \rightarrow \\
\vec{n}(c^{\uparrow}_{0002}) & \rightarrow \\
\vec{n}(c^{\uparrow}_{2222}) & \rightarrow \\
\vec{n}(c^{\uparrow}_{0222}) & \rightarrow
\end{matrix}
\begin{bmatrix}
1 & 0 & 4 & 0 & 0 \\
0 & 1 & 3 & 1 & 0 \\
1 & 0 & 3 & 0 & 1 \\
1 & 0 & 0 & 0 & 4 \\
1 & 0 & 1 & 0 & 3
\end{bmatrix}
\cdot
\begin{bmatrix}
u^{\uparrow} \\
u^{\downarrow} \\
v_0 \\
v_1 \\
v_2
\end{bmatrix}
=
\begin{bmatrix}
N \\
N \\
N \\
N \\
N
\end{bmatrix}.
\tag{11}
$$

If the tensor $c$ possesses an (approximate) $U(1)$ symmetry structure (as in the example above), then the linear system has a non-trivial solution in terms of charges $\vec{u}$ and $\vec{v}$. To solve it, it is known that one needs to bring the matrix $M$ into its Smith normal form (see Appendix A). Note here that the integer $N$ can, in fact, be changed arbitrarily. Although, the explicit values of the charges will depend on $N$, the $U(1)$ class of $a$ tensors will not. In other word, there is some "gauge" freedom to determine each $U(1)$ class. For the example with $D = 3$ considered here, we get integer charges, $u^{\uparrow} = +1$, $u^{\downarrow} = -1$, $v_0 = 0$, $v_1 = 2$ and $v_2 = 0$, as can be checked by direct substitution in Eq. (11) choosing $N = 1$. A complete list of the relevant charges are shown in Table 1 for bond dimension up to $D = 9$.

## 2.2 Reduced density matrices, CTM algorithm, and implementation details

The evaluation of energy is realized through two distinct reduced-density matrices (RDM), $\rho^{(NN)}$ and $\rho^{(NNN)}$, for nearest and next-nearest neighbour sites respectively. Their diagrammatic definition is shown in Fig. 1. The energy per site is then given by:

$$
e = 2J_1 \text{Tr}\left[\rho^{(NN)} \mathbf{S} \cdot \tilde{\mathbf{S}}\right] + 2J_2 \text{Tr}\left[\rho^{(NNN)} \mathbf{S} \cdot \mathbf{S}\right],
\tag{12}
$$

with $\tilde{S}^{\alpha} = -\sigma^y S^{\alpha}(\sigma^y)^T$, as these are the only non-equivalent terms of Hamiltonian (1) acting on the single-site iPEPS with $C_{4v}$ symmetry.

The two RDMs are obtained by substituting the environment of a $2 \times 2$ cluster within the infinite tensor network with the CTM approximation and tracing out all but two (nearest-neighbor or next-nearest-neighbor) sites. The leading computational cost in contraction of these networks is $O[(\chi D^2)^3 p^2]$ with $p = 2$ being the dimension of the physical index $s$. A more complete alternative is to consider a RDM of *all* four spins contained within the cluster. However, contracting such network with eight open physical indices is more expensive in terms of computational complexity and memory requirements, as both are amplified by a factor of $p^2$.

The most demanding part of the calculations is the CTM algorithm. Given the highly constrained nature of our iPEPS, in particular the $C_{4v}$ symmetry imposed on tensor $a$, we can utilize the efficient formulation of the algorithm of Ref. [33]. The $C_{4v}$ symmetry of the on-site tensor $a$ is reflected in the corner matrix $C$ which is taken to be diagonal and half-row-/-column tensor $T$ which is symmetric with respect to the permutation of its environment indices. We show the diagrammatic description of the main steps within single CTM iteration in the Fig. 2.

There are few more remarks to be made regarding the implementation of the CTM algorithm. The initial $C$ and $T$ tensors are given by partially contracted double-layer tensor, e.g. $C_{(dd')(rr')} = \sum_{sul} a^s_{uldr} a^s_{uld'r'}$. In addition, after each step of the CTM the tensors $C$ and $T$ are symmetrized accordingly and normalized by their largest element. To establish the convergence of the CTM, we use the RDM of nearest neighbors $\rho^{(NN)}_{2\times1}$ computed just from the $2 \times 1$ cluster at each CTM step. Once the difference (in sense of Frobenius norm) between $\rho^{(NN)}_{2\times1}$ from two consecutive iterations becomes smaller than $\epsilon_{CTM}$, we consider the CTM converged. During optimization we set $\epsilon_{CTM} = 10^{-8}$, which typically requires at most $O(70)$ iterations to converge for largest $(D, \chi_{opt}) = (7, 147)$ simulations considered. For scaling of observables of

$$E = \quad \Rightarrow \quad E_\chi = \quad = \sum_{i=0}^{D^2\chi^2-1} \lambda_i |B_i\rangle\langle B_i|$$

$$\rho_\chi^{(2)}(r) = \quad \cdots \left( \quad \right)^{(r-1)} \cdots$$

Figure 3: Top: Definition of the transfer matrix $E$ and its finite-$\chi$ approximation $E_\chi$ given by the converged $T$ tensor. Due to $C_{4v}$ symmetry imposed on the ansatz, the transfer matrix is symmetric and can be diagonalized. Eigenvalues are ordered with descending magnitude with the leading eigenvalue $\lambda_0$ normalized to unity. Bottom: RDM for two-point correlation functions, defined for $r \geq 1$, and its connection to transfer matrix $E$.

optimized states we instead iterate CTM until $\epsilon_{CTM} = 10^{-12}$. Remarkably, the $U(1)$ symmetry is preserved along the CTM procedure, whenever we adjust the truncation as to never break the multiplet structure of the enlarged corner.

Finally, a peculiar complication is present in the process of computing gradients by AAD, with two distinct aspects. First, the standard definition of adjoint function of eigenvalue (or singular value) decomposition relies on computing the full decomposition [43]. Hence, in this context one cannot resort to significantly faster partial decompositions such as Lanczos (at least during gradient computation). This sets the leading complexity of CTM iteration to $O[(\chi D^2)^3]$. Recently, a developed differentiable dominant eigensolver tries to address this shortcoming by alternative adjoint formula [44]. The second, more fundamental aspect is the ill-defined adjoint in the case of degenerate eigenvalues stemming from the terms proportional to the inverse of spectral gaps. We use a smooth cutoff function [39] to tame this problematic terms. Although doing so, the accidental crossings of eigenvalues in course of CTM sometimes result in erroneous gradients. In general, we found this occurrence, manifested by the failure of linesearch, to be rare. The formulation of AAD applied to gauge-invariant scalars (such as energy), whose computation however involves eigendecomposition with degenerate spectrum, still remains an open problem.

The complete algorithm is available as a part of the open-source library *peps-torch* [45] focused on AAD optimization of iPEPS.

## 3 Results

Our analysis is based upon an extensive set of calculations for various bond dimensions, ranging from $D = 2$ to 7, and different values of the frustrating ratio $J_2/J_1$ up to 0.5. For the large bond dimensions considered, the optimizations have been performed with environment dimensions up to $\chi_{opt} = 4D^2$ in the case of $D = 5, 6$ and up to $\chi_{opt} = 3D^2$ for $D = 7$. Here, we

want to highlight a few important aspects of iPEPS that are crucial for the investigation of the magnetically ordered phase. First of all, within optimizations with no imposed symmetries, there is a generic tendency to break the physical $U(1)$ symmetry of the Néel state (corresponding to global rotations around the axis of the spontaneous magnetization), leading to a slight (spin) nematic order, e.g., different values of the nearest-neighbour $S^x S^x$ and $S^y S^y$ correlations. This effect becomes more severe with increased frustration. For example, for most of the states with $D > 3$ and $J_2 \gtrsim 0.3$, there is a sensible (e.g., $5-10\%$ and even larger) difference in the correlation lengths corresponding to the transverse directions. Connected to this issue, we observe that it is possible to stabilize distinct "families" of local minima for various bond dimensions $D$, in particular $D = 3$ and $4$, with substantial differences in their magnetization, correlation length, and the degree of nematic order. Every family corresponds to a specific way the quantum fluctuations are built on top of the classical Néel state, e.g., by converging towards one of the possible choices of $U(1)$ charges or breaking the symmetry completely. Given the limited number of bond dimensions that are available within our AAD optimization, it is then of utmost importance to identify the family of minima that are connected and lead to a smooth and physically sound extrapolation in the $D \to \infty$ limit. Therefore, using the scheme introduced in Sec. 2.1, we take the optimized and almost $U(1)$-symmetric states from unrestricted simulations (typically for $J_2 \approx 0$) and infer their charge structure. The charges revealed by this analysis are listed in Table 1 and define the correct classes of $C_{4v}$-symmetric $U(1)$ iPEPS for $D$ ranging from 2 to 7, which best describe the Néel phase.

In order to obtain the thermodynamic estimates of the ground-state energy and magnetization (within the magnetically ordered phase), we compute these quantities for increasing values of the bond dimension $D$. A brute-force extrapolation in $1/D$ provides poor estimates, given the fact that the data are usually scattered, see for example the case of the magnetization reported in Appendix B. Instead, we follow the recent proposal that has been put forward in Refs. [41, 42]. In this respect, for every value of $D$ used, we compute the dominant correlation length $\xi$ which is defined by the so-called transfer matrix $E$ of iPEPS, see Fig. 3:

$$\xi = -\frac{1}{\log|\lambda_1|}, \tag{13}$$

where $\lambda_1$ is the second largest eigenvalue of the transfer matrix (without the loss of generality we assume that the largest one is normalized to 1). We remark that the value of $\xi$ obtained in this way coincides with the correlation length of the usual spin-spin correlation function (or, more precisely, the transverse correlations):

$$\langle \mathbf{S}_0 \cdot \mathbf{S}_r \rangle = \begin{cases} \mathrm{Tr}[\rho^{(2)}(r)\mathbf{S} \cdot \mathbf{S}] & r \in \text{even} \\ \mathrm{Tr}[\rho^{(2)}(r)\mathbf{S} \cdot \tilde{\mathbf{S}}] & r \in \text{odd} \end{cases}, \tag{14}$$

where $\rho^{(2)}(r)$, defined in Fig. 3, is the two-point RDM. To obtain the $\chi \to \infty$ limit of the correlation length, we use the scaling formula [41, 46]:

$$\frac{1}{\xi(\chi)} = \frac{1}{\xi(\infty)} + \alpha \left( \log \left| \frac{\lambda_3(\chi)}{\lambda_1(\chi)} \right| \right)^{\beta}, \tag{15}$$

which allows for more precise extrapolation of $\xi$ than the usual $1/\chi$ scaling across all ratios of $J_2/J_1$ [1]. Finally, the thermodynamic estimates of the energy and magnetization (squared)

---

[1] In general one uses ratio of the second and third largest eigenvalues, $\lambda_1$ and $\lambda_2$; however, due to $U(1)$ symmetry, they are always degenerate, forcing us to consider the next largest eigenvalue $\lambda_3$.

Table 2: Ground-state energies (in units of $J_1$) $e(D,\chi)$ and magnetization square $m^2(D,\chi)$ for $D = 7$, which can be considered as upper bounds of the exact $D \to \infty$ values. The tensor was optimized up to an environment dimension $\chi_{opt} = 3D^2 = 147$. The $\chi \to \infty$ extrapolations are done from environment bond dimensions $\chi \in [D^2, 13D^2]$.

| $J_2/J_1$ | $e(7,147)$ | $e(7,\chi \to \infty)$ | $m^2(7,147)$ | $m^2(7,\chi \to \infty)$ |
|---|---|---|---|---|
| 0.0 | -0.669428 | -0.669432 | 0.0994 | 0.0994 |
| 0.05 | -0.649273 | -0.649277 | 0.0926 | 0.0926 |
| 0.1 | -0.629497 | -0.629501 | 0.0852 | 0.0852 |
| 0.15 | -0.610154 | -0.610159 | 0.0771 | 0.0771 |
| 0.2 | -0.591314 | -0.591320 | 0.0685 | 0.0685 |
| 0.25 | -0.573067 | -0.573076 | 0.0591 | 0.0591 |
| 0.3 | -0.555520 | -0.555533 | 0.0491 | 0.0491 |
| 0.35 | -0.538850 | -0.538867 | 0.0383 | 0.0382 |
| 0.4 | -0.523054 | -0.523259 | 0.0270 | 0.0268 |
| 0.45 | -0.508895 | -0.508976 | 0.0173 | 0.0173 |
| 0.5 | -0.496152 | -0.496289 | 0.0086 | 0.0086 |

are obtained by a suitable fit in powers of $1/\xi$:

$$e(\xi) = e(\infty) + \frac{A}{\xi^3} + O\left(\frac{1}{\xi^4}\right), \tag{16}$$

$$m^2(\xi) = m^2(\infty) + \frac{B}{\xi} + O\left(\frac{1}{\xi^2}\right), \tag{17}$$

where $m = |\text{Tr}[\rho^{(1)}\mathbf{S}]|$ and $\rho^{(1)}$ is the single site RDM.

Let us start discussing the ground-state energy, shown in Fig. 4. For the unfrustrated case $J_2 = 0$, our results are fully compatible with what has been previously obtained in Refs. [41, 42]. The data points align perfectly according to the theoretical expectations and the extrapolated values are in very good agreement with quantum Monte Carlo results [4,5]. For example for $D = 7$ (after extrapolation in the environment dimension $\chi$) we get $e(D = 7) = -0.669432$, which is identical to the linear extrapolation in $1/\xi^3$ from $D = 3$ to 7. Including the subleading term $1/\xi^4$, the extrapolation gives $e(\infty) = -0.669437(2)$ (to be compared with the exact value $e_{\text{QMC}} = -0.669437(5)$ [4]).

For future comparisons, the energies for $D = 7$ and different $J_2/J_1$ ratios are reported in Table 2. Under increasing the frustrating ratio, a remarkably smooth behavior persists up to $J_2/J_1 \approx 0.3$; then, for larger values, small fluctuations on the fourth digit of the energy, are visible, possibily indicating that the scaling regime moves to larger values of $\xi$ (or $D$), not reachable within our current possibilities. Still, the quality of the results is sufficient to obtain reliable extrapolations for $\xi \to \infty$. Our calculations show that the expected scaling is not limited to the unfrustrated case, but persists in the whole antiferromagnetic region, thus corroborating the ideas put forward in Refs. [41,42]. One remarkable feature is that, while for small values of $D$ (i.e., for $D = 2$ and 3), the correlation length $\xi$ clearly *increases* by increasing $J_2/J_1$, for larger values of $D$ (i.e., for $D = 4$, 5, 6, and 7), it is essentially constant, or even slightly decreasing with $J_2/J_1$. This aspect will be discussed in connection to the magnetization curve that is presented below.

Then, we move to the central part of the present work, which deals with the magnetization, see Fig. 5. Here, we report $m^2(\xi)$ for different values of $J_2/J_1$ (including 0.5) for $D$ ranging





**Figure 4:** Finite correlation-length scaling of the energy per site for the $C_{4v}$-symmetric $U(1)$ iPEPS *Ansatz* with bond dimensions $D = 3, \ldots, 7$ (denoted by triangles, hexagons, pluses, diamonds, and crosses in the same order). Continuous lines are linear fits in $1/\xi^3$ which is the expected scaling in the magnetically ordered phase [41].

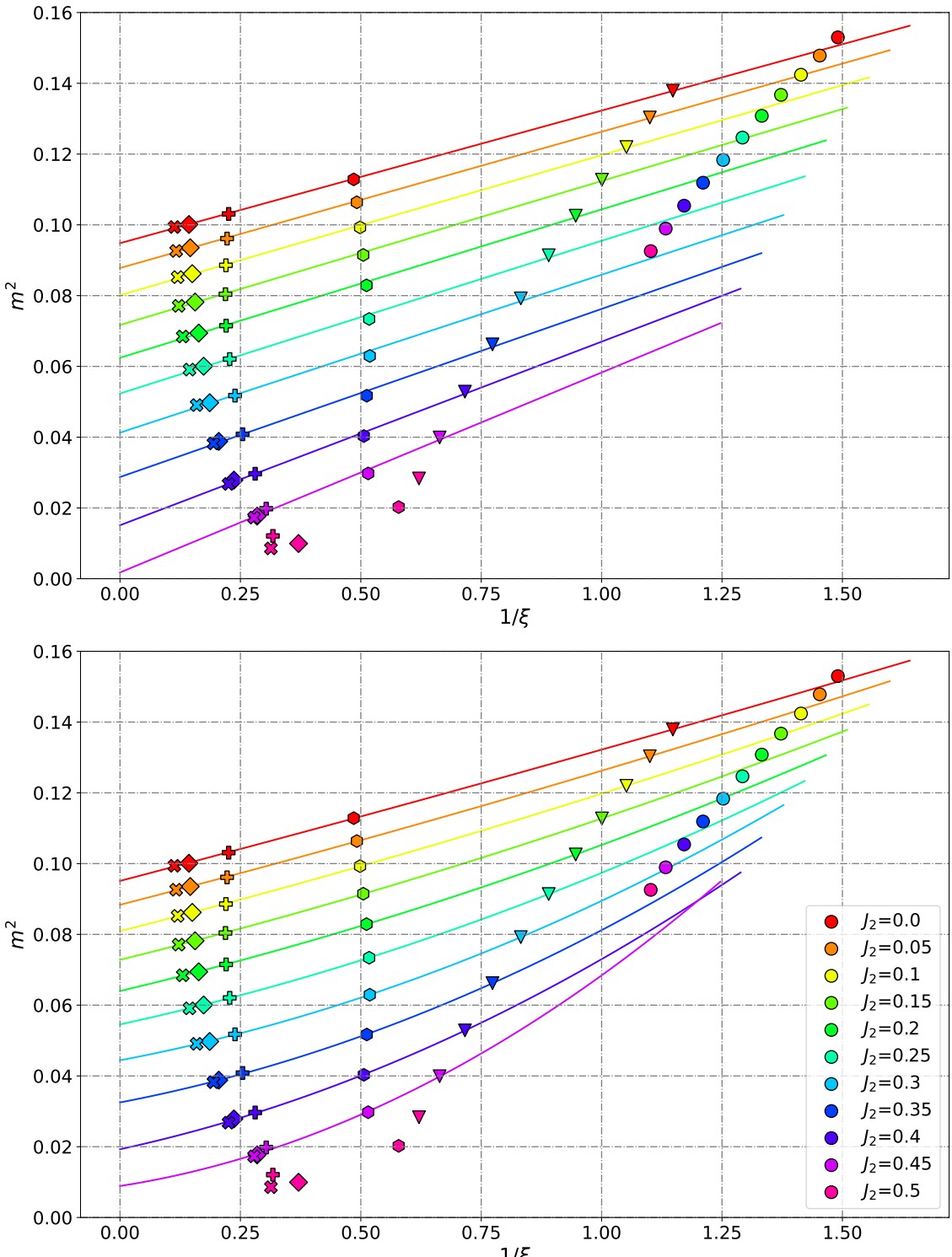

Figure 5: Finite correlation-length scaling of the magnetization for the $C_{4v}$-symmetric $U(1)$ iPEPS *Ansatz* with bond dimensions $D = 2,\ldots,7$ (denoted by circles, triangles, hexagons, pluses, diamonds, and crosses in the same order). The magnetization is plotted as a function of $1/\xi$, expected in the magnetically ordered phase [41]. Linear (quadratic) extrapolations of magnetization, excluding $D = 2$ data, are reported in the top (bottom) panel, except for $J_2/J_1 = 0.5$.

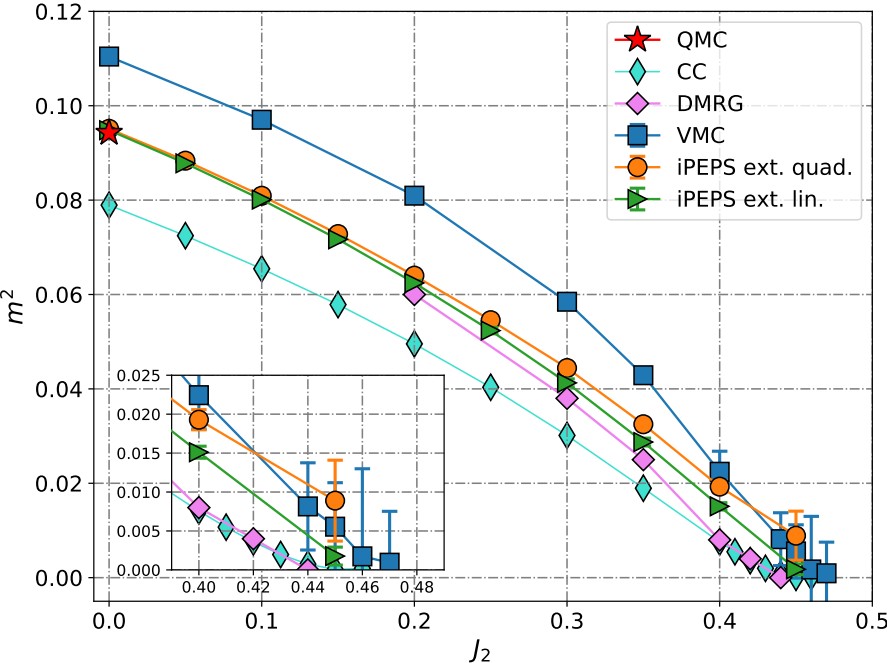

Figure 6: Magnetization (square) as a function of the frustrating ratio $J_2/J_1$ as obtained from Fig. 5. The exact result for $J_2 = 0$ is shown [4]. For comparison, the data from coupled-cluster [13], DMRG [17], and VMC [47] are also included. The inset shows the region $0.39 < J_2/J_1 < 0.49$.

from 2 to 7. Furthermore, the raw data for $D = 7$ are also shown in Table 2. In the unfrustrated case, we get $m^2(D = 7) = 0.0994$ and $m^2(\infty) = 0.0948(2)$, to be compared with the exact value $m^2_{QMC} = 0.0942(2)$ [4]. In Fig. 5, we attempt both linear and quadratic fits. As in the case of energy extrapolations, we exclude the results with $D = 2$ from the fitting procedure, since they are clearly off, especially for intermediate and large values of $J_2/J_1$. According to our fits, the linear one looks more trustable than the quadratic one, which serves to give an upperbound to the value of the magnetization. Within the linear fit, we observe vanishing magnetization for $J_2/J_1 \approx 0.46(1)$, giving rise to a continuous transition to a magnetically disordered phase, whose nature is beyond the scope of the present work. We would like to emphasize that the results for $J_2/J_1 = 0.5$ are clearly incompatible with a smooth behavior in $1/\xi$, strongly suggesting that at this point the ground state is already outside the magnetically-ordered phase. The final magnetization curve is shown in Fig. 6.

For comparison, the results obtained by coupled-cluster approximation [13], DMRG [17], and VMC [47] (based on Gutzwiller-projected fermionic states) are also shown. In the latter case, a quantum critical point for $J_2/J_1 \approx 0.48$, separating the antiferromagnetic phase and a gapless spin liquid, has been reported. The present results are expected to improve the accuracy of the magnetization (e.g., the accuracy of $m^2$ for the unfrustated case is smaller than 1%). Still, these two independent calculations give very similar behavior, with almost compatible values for the location of the quantum critical point. We would like to mention that, recent numerical calculations, including DMRG [18], neural-network approaches (based upon restricted Boltzmann machines on top of fermionic states) [23], and finite size PEPS calculations [48] also pointed out that the Néel phase survives up to $J_2/J_1$ in the range $0.45 \div 0.47$, a value that is considerably larger than the one predicted by linear spin-wave theory [49].

Finally, we would like to comment on the $J_2$-dependence of the correlation length, which is clearly different at small (i.e. $D = 2, 3$) and larger (i.e. $D = 4, \cdots, 7$) bond dimensions.

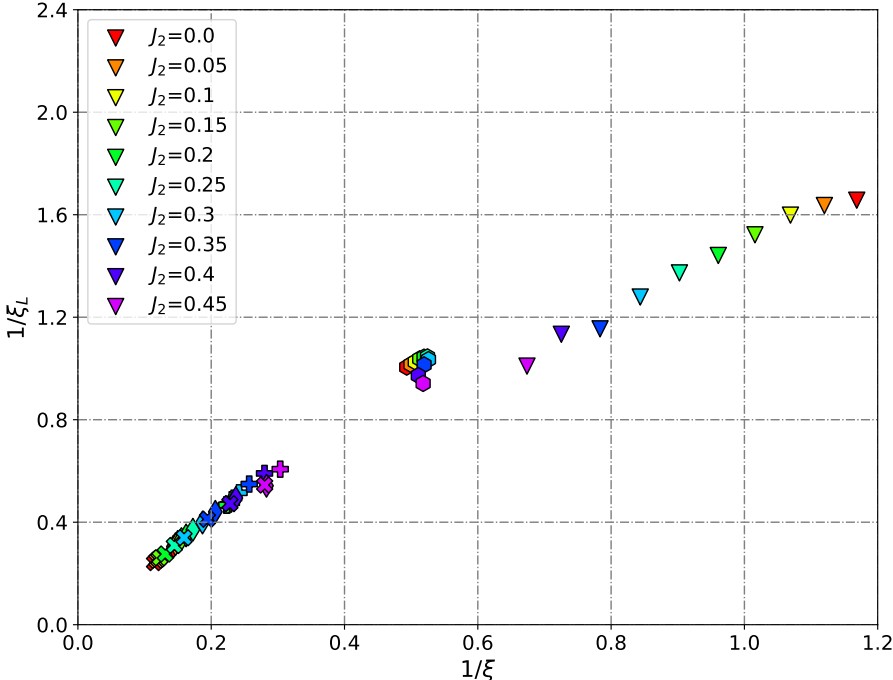

Figure 7: Longitudinal correlation length $\xi_L$, as extracted from the spin-spin correlations, as a function of the transverse one $\xi$ for different values of $J_2/J_1$ at $D = 3, \dots, 7$ (denoted by triangles, hexagons, pluses, diamonds, and crosses in the same order).

A possible explanation of the rapid increase of $\xi$, for $D = 2$ and 3, when approaching the critical point, may be attributed to the fact that, for these very small bond dimensions, the antiferromagnetic state is poorly approximated as a "dressed" product state, having a finite magnetization but lacking the correct transverse (Goldstone) fluctuations. When approaching the phase transition, the magnetization decreases and the state starts to build up long-range entanglement (for $D = 3$ a short-range resonating-valence bond state can be constructed [50]). Therefore, a larger correlation length can be attained. Once the basic (low $D$) structure of tensor $a$ is established, optimizing at increasingly higher $D$ further improves the description of the antiferromagnetic state and allows correlation length to grow, becoming large even in the presence of significant frustration. Then, no appreciable change of $\xi$ is detected when approaching the quantum critical point. In this respect, we expect that $\xi \to \infty$ in the whole Néel phase, including the critical point. Remarkably, despite optimized iPEPS being finitely correlated, the correct exponent of the power-law decay of transverse spin-spin correlations, i.e., $\langle S_0^x S_r^x \rangle \simeq 1/r$ (assuming magnetization along $z$-spin axis), can already be obtained, see Appendix C for the case with $J_2 = 0$. As mentioned above, $\xi$ corresponds to the correlation length of transverse spin-spin correlations. In addition to that, it is possible to evaluate, by a direct fitting procedure of the correlation function itself, the correlation length $\xi_L$ of the longitudinal correlations. We find also this quantity to be relatively large, i.e., $\xi_L \approx \xi/2$, see Fig. 7. Moreover, as for transverse spin-spin correlations, the short-range behavior of the longitudinal correlations reveals their power-law decay (see Appendix C), which then becomes rapidly cut off above the finite-$D$ induced length scale $\xi_L$. These findings show that our optimized iPEPS are even able to approximately capture the power-law behavior of transverse and longitudinal spin-spin correlations of the Néel phase.

# 4 Conclusions

In this work, we have investigated the antiferromagnetic phase of the spin-1/2 $J_1 - J_2$ model on the square lattice, evaluating with unprecedented accuracy the energies and magnetizations for $J_2/J_1 \leq 0.45$. The results point towards the existence of a quantum critical point at $J_2/J_1 \approx 0.46(1)$, which separate the Néel antiferromagnet and a quantum paramagnet, whose nature is beyond the scope of the present study. From the methodological side, we combined state-of-the-art optimization techniques (based upon the AAD scheme [39]), clever parametrizations of the tensor network (based upon the underlying residual $U(1)$ symmetry that exists in the Néel phase), and recently developed extrapolation analyses (based upon the correlation-length scaling [41, 42]). In particular, the construction of $U(1)$-symmetric tensor is pivotal to a straight optimization procedure and correlation-length scaling to solid extrapolations to thermodynamic limit. With these tools in hand, it is possible to get reliable estimations for the ground-state energy but, most importantly, also for the magnetization within the frustrated regime, for which no exact methods can be applied. Therefore, the main outcome of the present work is to provide the magnetization curve for the spin-1/2 $J_1 - J_2$ model on the square lattice up to relatively large values of the frustrating ratios. In particular, the magnetization curve shows a smooth behavior, which strongly suggest the existence of a continuous phase transition towards a quantum paramagnet.

Here, our calculations have been limited to the magnetically ordered phase, where relatively entangled states have been achieved. Indeed, rather long correlation lengths are obtained, indication that the tensor network may approximately describe the existence of gapless excitations (i.e., Goldstone modes). The magnetically disordered phase still remains elusive, most probably because of its high-entangled nature due to fractional excitations (spinons and visons). Although the present optimization of $U(1)$-symmetric iPEPS can be readily extended beyond $J_2 > 0.45$, there is a priori no justification for the choice of $U(1)$ charges as inferred from the Néel phase to also describe the best variational states of the disordered phase. Moreover, due to the tendency of unrestricted optimizations to develop nematic order at large values of frustration one cannot extract the relevant $U(1)$ charges from such states. In this respect, the recently-developed method to directly impose $SU(2)$ symmetry [36, 51] would be beneficial to the final understanding of the full phase diagram of the spin-1/2 $J_1 - J_2$ model.

## Acknowledgements

We thank Fabien Alet, Zhengcheng Gu, Andreas Läuchli, Wenyuan Liu, Pierre Pujol, Anders Sandvik, and Sandro Sorella for helpful discussions. We thank Shoushu Gong and Johannes Richter for providing the DMRG and coupled-cluster data.

**Funding information**   This work was granted access to the HPC resources of CALMIP supercomputing center under the allocation 2020-P1231. This project is supported by the TNSTRONG ANR-16-CE30-0025 and the TNTOP ANR-18-CE30-0026-01 grants awarded by the French Research Council.

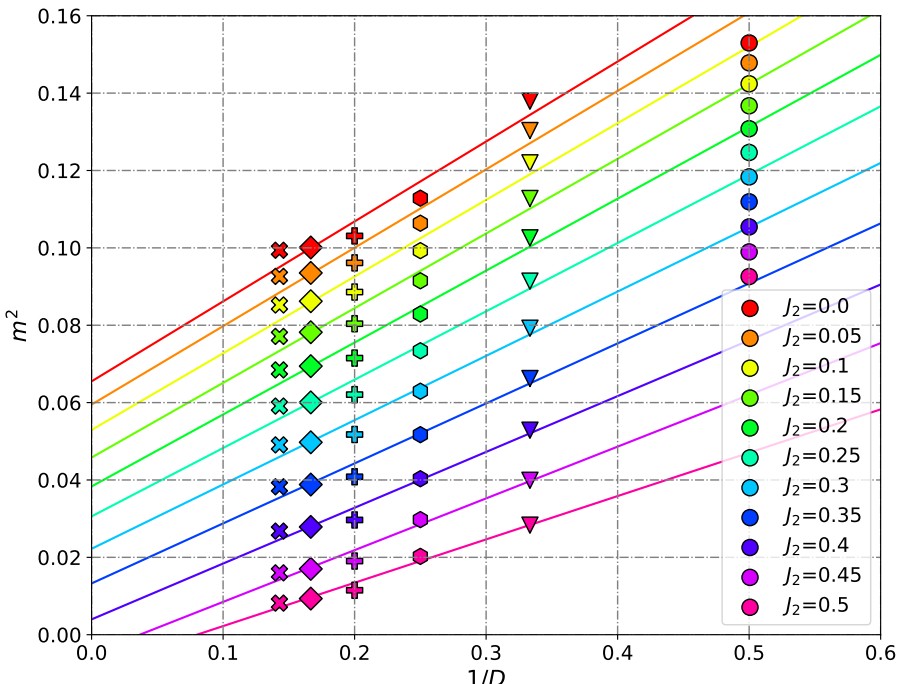

Figure 8: Linear extrapolation in $1/D$ for the $C_{4v}$-symmetric $U(1)$ iPEPS Ansatz with bond dimensions $D = 2, \ldots, 7$ denoted by circles, triangles, hexagons, pluses, diamonds, and crosses in the same order ($D = 2$ data is excluded from the fit). Data is the same as in Fig. 5.

## A  Smith normal form

The Smith normal form of matrix $M$ is needed to solve the linear system introduced in Sec. 2.1. For a $n \times m$ integer matrix $M$ the Smith normal form is defined as

$$LMR = S, \tag{18}$$

with $L$ and $R$ being integer matrices with unit determinant and $n \times m$ integer matrix $S$. The only non-zero elements of $S$ are $S_{i,j} = s_i \delta_{i,i}$ for $1 \leq i \leq r$ where $r \leq m$. These so-called *invariant factors* $s_i$ satisfy divisibility relations $s_i | s_{i+1}$ for $1 \leq i < r$. The Smith Normal form conveniently reveals the vectors of integer charges ($\vec{u}$ and $\vec{v}$) spanning the $m - r$ dimensional kernel of the constraint system $M$ as the last $m - r$ columns of matrix $R$. Let us remark that such kernel vectors are unique up to an arbitrary multiples of trivial charge vectors $\vec{K}_0 = [1, 1, 0, \ldots, 0]$ and $\vec{K}_1 = [0, 0, 1, \ldots, 1]$, as these merely move the constant $N$. In detail, a set of tensor elements $a^s_{uldr}$ satisfying $M \cdot (\vec{u}, \vec{v}) = 0$ is identical to the set of elements satisfying $M \cdot [(\vec{u}, \vec{v}) + \alpha \vec{K}_0 + \beta \vec{K}_1] = \alpha + 4\beta$ with $\alpha, \beta \in \mathbb{Z}$.

## B  $1/D$ extrapolations

In Fig. 8, we report the results of the magnetization as a function of $1/D$. In this case, the magnetization cannot be described by a simple linear function in $1/D$ with appreciable accuracy. Considerable deviations are present, especially for larger bond dimensions across the studied range of $J_2/J_1$, preventing a smooth extrapolation in the $D \to \infty$ limit.

## C Spin-spin correlations in the $J_2 = 0$ limit

In Fig. 9, assuming magnetization along $z$-spin axis, we show the decay of both transverse $\langle S_0^x S_r^x \rangle$ and longitudinal $\langle S_0^z S_r^z \rangle$ correlations for $J_2 = 0$ and $D = 2, \ldots, 7$. Due to imposed $U(1)$ symmetry the transverse correlations along $x$ and $y$ spin axes, $\langle S_0^x S_r^x \rangle$ and $\langle S_0^y S_r^y \rangle$, are identical. The extrapolated values are obtained by performing, for each distance $r$, an extrapolation in $1/\xi$ of the finite-$D$ results using the three largest available bond dimensions $D = 5, 6$, and 7. Then, the extrapolated correlations are fitted in the short-distance region $r \in [2, 11]$ (excluding the nearest-neighbor case) with a power law $f(r) \propto r^{-\beta}$. The final result gives an exponent $\beta \approx 1.02(1)$ for transverse and $\beta_L \approx 1.90(5)$ for longitudinal correlations.

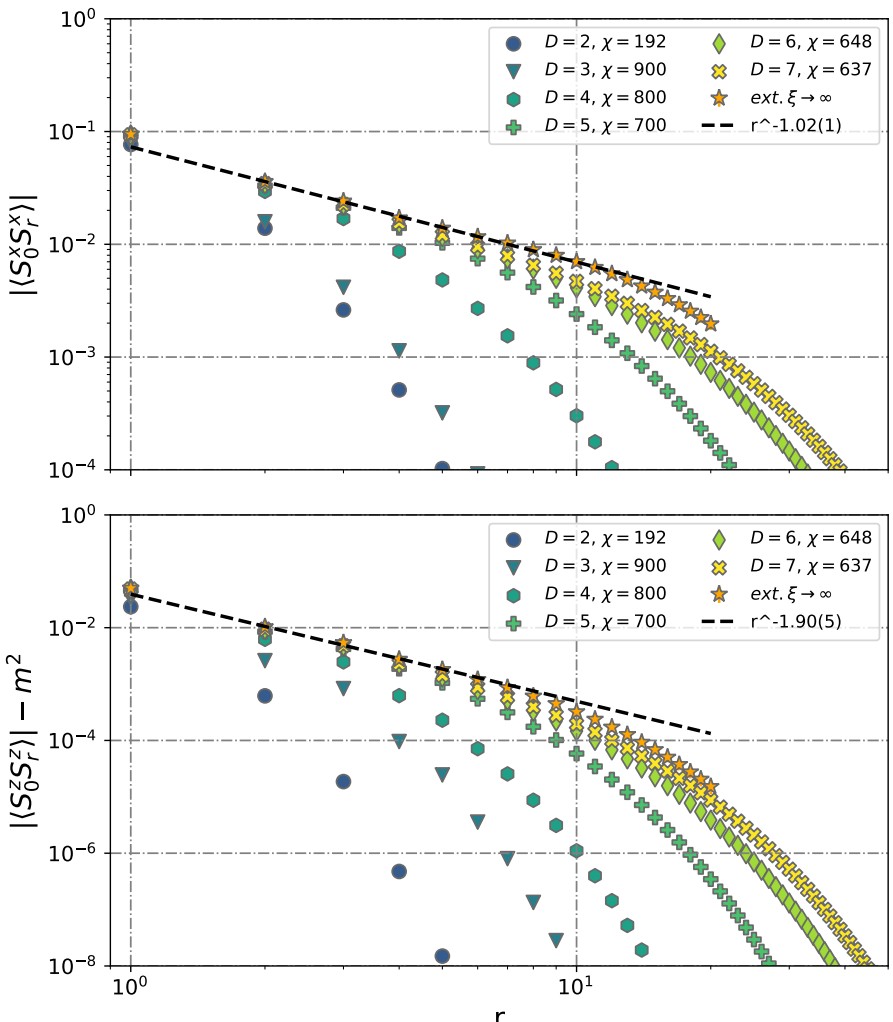

Figure 9: Transverse (longitudinal) spin-spin correlations at $J_2 = 0$ are shown in the upper (lower) panel, for $D = 2, \ldots, 7$. Linear extrapolations in $1/\xi$, up to $r = 20$, are performed using the $D = 5, 6, 7$ data. The dashed lines are power-law fits to short-distance behavior, see text.

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
