# Peer review of "Investigation of the Néel phase of the frustrated Heisenberg antiferromagnet by differentiable symmetric tensor networks"

_SciPost Physics, doi:SciPost Phys. 10, 012 (2021)_

## Round 1 · Referee Report · Anonymous (Referee 1) · 2020-12-7

Report

In this paper the authors study the ground state of the spin-half
J1-J2 model on the square lattice by means of differentiable
symmetric tensor networks.
This model is a paradigmatic highly frustrated quantum model and
it allows the investigation of frustration driven quantum phase
transitions between semi-classical ordered and quantum
paramagnetic phases.
Meanwhile there are numerous studies of this model and there is
some consensus on the ground state phase diagram albeit the
discussion of the nature of the quantum paramagnetic phase is
still ongoing.
The model may also serve as a test ground for new quantum many
body methods, such as the tensor network approach.
The focus of the present work is on the methodology of
differentiable symmetric tensor networks, where the J1-J2
model serves as a challenging problem of frustrated quantum
magnetism to demonstrate the potential of the method.
On the other hand, the main result, namely the dependence of the order Neel parameter on the frustrating coupling does not provide new
insight in the physics of the model, rather it confirms the
well-known second-order transition at J2/J2 ~ 0.45.

All in all, I felt that this paper provides
a valuable contribution to the field which deserves publication.

I have several concerns, however, that should be addressed before I
can recommend this paper for publication.

(1) Bearing in mind the long history of studies of the J1-J2 model
it is useful to mention and briefly discuss other (except tensor
network and related studies) successful approaches
such as high order series expansion, PRB 60, 7278 (1999) and
PRB 73, 184420 (2006);
coupled cluster approach, PRB 78, 214415 (2008), Eur.Phys.J.B 88, 2 (2015);
cluster mean-field theory, J. Phys Condens. Mat. 26, 115601 (2014).

(2)
In Fig.6 the authors compare their order parameter with QMC data at J2=0 and their previous results of Ref. 42.
It could be useful to compare also with order-parameter data
from DMRG, Ref 14 (PRL 113, 027201 (2014)), and coupled cluster
approach.

(3) For me it remains unclear why the authors do not discuss the
region J2/J1 > 0.5.
I believe it would be useful to add a brief discussion to what
extent the used method is appropriate to investigate the gap in the
quantum paramagnetic phase as well as the transition
quantum paramagnet and the semi-classical ordered striped phase
at J2/J1 ~ 0.6, which is most likely of 1st order.
  • validity: -
  • significance: -
  • originality: -
  • clarity: -
  • formatting: -
  • grammar: -

Author:  Juraj Hasik  on 2020-12-31  [id 1122]

(in reply to Report 1 on 2020-12-07)
Category:
answer to question
reply to objection
pointer to related literature

Dear Referee,

thank for your assessment of our manuscript and your feedback. Let us adress your comments:

(1) Bearing in mind the long history of studies of the J1-J2 model it is useful to mention and briefly discuss other (except tensor network and related studies) successful approaches such as high order series expansion, PRB 60, 7278 (1999) and PRB 73, 184420 (2006); coupled cluster approach, PRB 78, 214415 (2008), Eur.Phys.J.B 88, 2 (2015); cluster mean-field theory, J. Phys Condens. Mat. 26, 115601 (2014).

We have added these references to the introduction.

(2) In Fig.6 the authors compare their order parameter with QMC data at J2=0 and their previous results of Ref. 42. It could be useful to compare also with order-parameter data from DMRG, Ref 14 (PRL 113, 027201 (2014)), and coupled cluster approach.

The data from DMRG of Ref 14 and coupled-cluster of Eur.Phys.J.B 88, 2 (2015) are now included in Fig. 6 for comparison.

(3) For me it remains unclear why the authors do not discuss the region J2/J1 > 0.5. I believe it would be useful to add a brief discussion to what extent the used method is appropriate to investigate the gap in the quantum paramagnetic phase as well as the transition quantum paramagnet and the semi-classical ordered striped phase at J2/J1 ~ 0.6, which is most likely of 1st order.

In principle, the single-site U(1)-symmetric iPEPS ansatz + gradient optimization with automatic differentiation can be readily used to study ground state beyond J2>0.5. However, a question arises: what are the U(1) charges that best describe the paramagnetic phase ? A priori there is no justification for U(1) charges which best describe the Neel phase (listed in table 1) to be also appropriate for the paramagnetic phase. The unrestricted (no U(1) symmetry) optimization in highly-frustrated region leads to states with nematic order and thus we cannot use such states to infer U(1) charges in an unbiased manner. Therefore, in the present work we have limited ourselves to the Neel phase for which we can reliably select the U(1) charges. To assess the paramagnetic phase, one has to first look for an alternative ways of choosing U(1) charges. We have included this discussion in the conclusions.

Assuming one has precise variational states in the paramagnetic region, the gap in the context of iPEPS is accessible by the means of recent advances in simulating excited states with iPEPS, see Phys. Rev. B 99, 165121 (2019) and Phys. Rev. B 101, 195109 (2020).

The transition between paramagnet and striped phase brings different complications. In particular one has to abandon single-site iPEPS ansatz and consider a unit-cell with more tensors as has been done in the Ref. 37.

---

## Round 1 · Referee Report · Anonymous (Referee 3) · 2020-12-8

Report

In the paper "Investigation of the Néel phase of the frustrated Heisenberg antiferromagnet by differentiable symmetric tensor networs" the authors present an estimation of the magnetization curve of the $J_1-J_2$ Heisenberg antiferromagnet on a square lattice.

They report that their $J_2$ at which the phase transition is located agrees with the range given in the literature.

The main results of this paper are the challenging methodological combination of previously described techniques and the resulting magnetization curve for the model at hand.

Both parts are presented in detail and the results can be reproduced easily (due to the fact that the used code is open source).

I believe this work is a notable contribution, especially for benchmarking future developments. Nevertheless I would suggest several (mainly cosmetic and hence debatable) improvements in the presentation prior to publication.

Requested changes

Consider giving (review) references to the mentioned methods (DMRG, fRG, VMC, MPS).

Please move the floating objects closer to their appearance in the text. Table 1 (page 7) is the first floating environment mentioned in the text on page 8, which is fine. But figure 1 (page 5) and figure 2 (page 6) are mentioned afterwards.

In figure 5 basically the same data is shown twice. Maybe consider two different line types for the different extrapolations within only one figure. Especially, since the quadratic extrapolation is supposed to be less trustworthy.

Consider adding an inset in figure 6 for the most interesting part ($J_2=0.4 - 0.5$).

As the error bars do not add that much insight (except in figure 6) they can as well be disabled.

Maybe consider to move figure 4 (or a portion of it) to an appendix, since only the first subfigure ($J_2=0$) is discussed in detail and the main result of the figure is summarized in table 2.

"Once the basic (low D) structure of tensor is established [...]", probably an identifier for the tensor is missing?

"The importance of our findings is twofold." Linguistically only one argument follows.

  • validity: high
  • significance: high
  • originality: good
  • clarity: high
  • formatting: good
  • grammar: good

Author:  Juraj Hasik  on 2020-12-31  [id 1121]

(in reply to Report 3 on 2020-12-08)
Category:
reply to objection
correction

Dear Referee,

first of all, thank you for careful reading of our manuscript and your feedback. Let us address your recommendations and comments:

Consider giving (review) references to the mentioned methods (DMRG, fRG, VMC, MPS).

We believe including the reviews is not necessary, instead their references can be found in the cited studies of J1-J2 model by the methods above.

Please move the floating objects closer to their appearance in the text. Table 1 (page 7) is the first floating environment >mentioned in the text on page 8, which is fine. But figure 1 (page 5) and figure 2 (page 6) are mentioned afterwards.

The floats in question have been repositioned to appear closer to the first appearance in the main text.

In figure 5 basically the same data is shown twice. Maybe consider two different line types for the different extrapolations >within only one figure. Especially, since the quadratic extrapolation is supposed to be less trustworthy.

We have experimented with several forms of presentation of this data. Merging two forms of extrapolation within single figure leads to overly dense plot. Since the space is not a limiting factor, we believe the current form provides the most clarity.

Consider adding an inset in figure 6 for the most interesting part (J2=0.4−0.5).

As per the request of 1st referee we have added additional data from DMRG and coupled-cluster studies to Fig. 6 and included the inset zooming in on the region where the Neel order vanishes.

As the error bars do not add that much insight (except in figure 6) they can as well be disabled.

The error bars have been omitted from all but figure 6 in the updated manuscript.

Maybe consider to move figure 4 (or a portion of it) to an appendix, since only the first subfigure (J2=0) is discussed in >detail and the main result of the figure is summarized in table 2.

Let us clarify, that the table 2 contains values of energy (and order parameter m^2) at finite bond dimension D=7 thus serving as an upper bound on the energy. Instead, the Figure 4 presents extrapolations to D->\infty by the means of correlation-length scaling. We believe Fig. 4 and Fig. 5 together give a comprehensive picture of correlation-length scaling analysis across the entire Neel phase and as such constitute integral part of the results and thus should stay in the main text.

"Once the basic (low D) structure of tensor is established [...]", probably an identifier for the tensor is missing?

Indeed, the sentence now reads: "[...] structure of tensor a is established [..]"

"The importance of our findings is twofold." Linguistically only one argument follows.

We have left out the sentence in question.

---

## Round 1 · Referee Report · Anonymous (Referee 2) · 2020-12-8

Report

Ground state properties of square-lattice S=1/2 Heisenberg model is studied by the variational state that is given by an infinite and uniform tensor contraction. The local tensor is expressed as the combination of linearly independent tensors, each of which satisfies the symmetry. This representation enables the systematic and stable numerical optimization of the variational state. The environment tensor is created by the CTMRG method, where the corner transfer matrix dimension is carefully chosen so that the cut-off effect does not affect the variational optimization process. Convergence in both the ground-state energy per site and magnetization is checked from D=2 to D=7. A remarkable point is that the correlation length of the optimized variational state is used for the final scaling analysis, and this physical scaling scheme realizes the good convergence. The technical explanations are well separated from the numerical data analysis, and therefore this article fits both to the experts in tensor network and also to general readers who are interested in 2D magnets. For these reasons, and to avoid disturbing the good structure of sentences, I recommend the publication of this article as it is, except for tiny corrections added by authors only if they are necessary.

---

## Round 2 · Referee Report · Anonymous (Referee 1) · 2021-1-2

Report

The authors have carefully addressed my concerns raised in the last
review and modified the manuscript accordingly.
I would recommend the work to be accepted for publication in SciPost Physics.

---

## Round 2 · Referee Report · Anonymous (Referee 3) · 2021-1-5

Report

The authors have addressed some of my suggestions of the last review. As the remaining ones are only of cosmetic nature, I recommend the work to be accepted for publication in SciPost Physics.

---

## Round 2 · Referee Report · Anonymous (Referee 2) · 2021-1-5

Strengths

Correlation length is used for the estimations toward large-D limit. This improves the convergence in spontaneous magnetization.

Weaknesses

Size of the unit cell limits the parameter range. The case J2 > J1 cannot be treated.

Report

Several rearrangements of figures and a table makes it easy to read through the article. Fine corrections based on comments from other referees are properly done. Again, I recommend the publication of this article.

---

## Round 2 · List of Changes

Additional references to works on J1-J2 model have been added to Introduction: Series expansions PRB 60, 7278 (1999), PRB 73, 184420 (2006); coupled cluster approach, PRB 78, 214415 (2008), Eur.Phys.J.B 88, 2 (2015); cluster mean-field theory, J. Phys Condens. Mat. 26, 115601 (2014).

The introduction of U(1) symmetry has been changed slightly. Formulas 6,7, and 8 now correctly include dependence
on the group element g \in U(1).

The data from DMRG of Ref 14 and coupled-cluster study of Eur.Phys.J.B 88, 2 (2015) have been added to Fig. 6 for comparison.
Also an inset zooming in on the region 0.39<J2<0.49 has been added.

A short discussion regarding the extension of the presented methodology beyond J2>0.5 has been added to Conclusions.

Data in all but Figure 6 is now presented without error bars. The figures have been repositioned to appear closer
to the point of their first mention in the main text.

Minor grammatical corrections of the text have been added.

---

## Editorial Decision

published